# A comparative study on the physical fitness of college students from different grades and majors in Jiangxi province

**Jianzhong Sun**[1,2], **Chan Lin**[1,2], **Lei Wang**[3], **Cunjian Bi**[1,2], **Bin Qiao**[3]*

**1** School of Physical Education, Chizhou University, Chizhou, China, **2** Sports Health Promotion Center, Chizhou University, Chizhou, China, **3** Basci Teaching Department, Chizhou Vocational and Technical College, Chizhou, China

* QB13305663823@126.com

## Abstract

### Objective

Research to date has not provided a clear understanding of how different grades and majors affect the physical fitness of college students. It is postulated that there are significant disparities in physical health among college students of different grades and majors. The purpose of this study was to evidence these health disparities and to engage in an extensive analysis and discussion thereof.

### Methods

A sample of 8,772 (2,404 boys and 6,368 girls) Chinese college students from freshman to junior years, aged 17–22, including 12 different majors in four colleges, were recruited in Jiangxi Province. All seven physical fitness indicators (body mass index (BMI), forced vital capacity, 50-m dash, standing long jump, sit and reach, upper body muscle strength, and endurance runs) were conducted for all participants. One-way ANOVA and LSD tests were conducted to compare the physical fitness scores of different grades in the same major. Independent sample *t*-tests were utilized to compare the differences in every physical fitness indicator for different majors. Pearson's correlations among 12 majors for every grade were conducted to study the significant corrections between the two physical fitness indicators. The body mass index (BMI) and physical fitness indicator (PFI) for college students of different grade were investigated using a nonlinear regression model.

### Results

The current state of physical fitness among college students is concerning, as the majority of students were barely passing (with a passing rate of 75.3%). Specifically, junior students exhibited lower scores than freshman and sophomore students across all 12 majors. From freshman to junior year, majors of music (78.01±4.58), English (79.29±5.03), and education (76.26±4.81) had the highest scores, respectively, but major art consistently scored the lowest, which were 73.85±6.02, 74.97±5.53, and 72.59±4.84, respectively. Pairwise

**Data Availability Statement:** The datasets used during the current study are publicly available at https://doi.org/10.5061/dryad.qbzkh18sd and can be found in supporting information.

**Funding:** This work were supported by "the Ministry of Education Humanities and Social Sciences Research Youth Project Approval of 2023 (Award No. 23YJC890002)", "School level scientific research project of Chizhou University in 2022 (Award No. CZ2022YJRC03)", and "Anhui province 2022 provincial quality engineering project for higher education institutions (Award No. 2022jxms116)". The funders had no role in study design, data collection and analysis, decision to publish, or preparation of the manuscript.

**Competing interests:** The authors have declared that no competing interests exist.

comparisons revealed more significant differences in individual physical fitness indicators among the three grades in humanities than in sciences. Pearson's correlations showed significant correlations among seven physical fitness indicators in all three grades. PFI had a parabolic trend with BMI both for boy and girl college students in Jiangxi province.

## Conclusion

The physical fitness indicators of college students in Jiangxi province significantly differed in grades and majors, showing undesirable phenomena. The physical fitness of senior and humanities major college students was much weaker and needs sufficient attention. The relationship between BMI and PFI presented an inverted "U"-shaped change characteristic. Continued nationwide interventions are needed to promote physical activity and other healthy lifestyle behaviors in China.

## Background

Physical inactivity can have significantly negative impacts on physical fitness, thereby increasing the risk of non-communicable diseases and mortality [1, 2]. In accordance with the latest World Health Organization (WHO) guidelines for physical activity, adults should aim to engage in at least 150–300 minutes of moderate intensity aerobic activity weekly, or 75–150 minutes of high-intensity aerobic activity weekly, or a combination of both moderate and high-intensity activities [3]. Most of Swedish children did not reach the recommended daily activity level [4]. Poor physical fitness leads to a decrease in the body's immune system, making it easier for bacteria and viruses to invade. Physical inactivity is a significant contributor to cardiovascular disease, diabetes, and obesity, which can heighten the risk of developing conditions such as hypertension, hyperlipidemia, osteoporosis, depression, diabetes, obesity, and heart disease, ultimately leading to life-threatening outcomes [5, 6]. Being overweight and obesity are also the main manifestations of poor physical fitness nowadays [7], and have been condemned as the fourth largest risk factors for death after hypertension, dietary risk, and tobacco use [8]. The latest data showed that over 1.3 million people worldwide die from overweight and obesity every year [9]. Poor lifestyle habits, such as physical inactivity and poor diets, also significantly impact physical fitness [10]. Su et al. revealed that total daily physical activity had a significant positive effect on cancer ($P < 0.05$) [11].

Fitness refers to a series of physical exercises designed to enhance muscle development, improve physical strength, enhance body shape, and improve mental well-being through the use of either bare hands or various equipment. It encompasses five key elements: clear fitness goals, a blend of aerobic and anaerobic activities, regular exercise, a balanced diet, and rest and recovery. According to the inaugural global health index, which evaluated 146 countries and regions, the average score was 46.96. China's score was 68.10, slightly above the global average [12].

There was a significant negative correlation between a sedentary lifestyle and physical fitness level ($P = 0.00$, $P < 0.05$) for Indonesian adolescents [13]. Cantarero et al. believed that the health quality of life was related to physical fitness and a sedentary lifestyle [14]. A significant positive relationship between leisure-time physical activity and physical fitness was demonstrated in both females and males aged 20–59 years [15]. The physical fitness of Chinese college students is also not encouraging. According to the results of national physical fitness

monitoring in 2020, the physical fitness of Chinese college students continues to decline [1, 16]. The main manifestation is that the body weight gradually increases, while strength and quality show a decreasing trend [17]. Physical fitness was shown to be positively associated with physical activity among adults aged 40 to 79 years in China [18]. Obesity, physical inactivity, and reduced physical fitness contribute to the rise of chronic diseases in China [19].

Contemporary college students are the future of our country, the hope of our nation, and shoulder the heavy responsibility of China's takeoff. The physical fitness and health of college students are related to the future and destiny of our country. To the best of our knowledge, there is currently no comprehensive research on the physical well-being of college students in Jiangxi Province, China. This study aims to serve as a valuable reference for enhancing the physical fitness of college students in Jiangxi province, China. Months ago, our research team conducted a cross-sectional study on age, sex, and body mass index (BMI) roles in physical fitness for Chinese college students [20, 21]. In this article, the different grades and majors were considered to reveal their impact on the physical fitness of Chinese college students. This can provide a more comprehensive understanding of the influencing factors of contemporary college students' physical fitness, provide guidance for universities to improve and enhance students' physical fitness [22] and provide data support for national sports educators to formulate policies [23, 24].

## Methods

### Data source and participants

In this study, four colleges and universities were randomly selected as test schools among all colleges and universities in Jiangxi Province, China. Executed according to the 2021 Chinese National Survey on Students' Constitution and Health (CNSSCH) [25]. This study mainly focuses on the impacts of grades and majors on the physical fitness of college students. Based on a multistage stratified random cluster sampling method, choose three grades from freshman to junior, including 12 majors (4 science majors and 8 humanities majors) from 4 colleges in Jiangxi province (Table 1). Finally, 8,772 (2,809 freshmen, 2,904 sophomores, and 3,059 juniors) college students were included in this study.

This study was approved by the human experimental ethics committee of Chizhou University (No. CZ2022YJRC03). Written informed consent was obtained from the participants

**Table 1. Participants of twelve majors in three grades of college students from Jiangxi, China.**

| Subject | Major | Freshman (%) | Sophomore (%) | Junior (%) | Total |
|---|---|---|---|---|---|
| Sciences | Math | 258 (9.2) | 278 (9.6) | 287 (9.4) | 823 |
| | Physics | 249 (8.9) | 261 (9.0) | 258 (8.4) | 768 |
| | Chemistry | 201 (7.2) | 264 (9.1) | 239 (7.8) | 704 |
| | Biology | 238 (8.5) | 275 (9.5) | 284 (9.3) | 797 |
| Humanities | Chinese | 235 (8.4) | 246 (8.5) | 260 (8.5) | 741 |
| | English | 262 (9.3) | 255 (8.8) | 249 (8.1) | 766 |
| | Law | 242 (8.6) | 223 (7.7) | 229 (7.5) | 694 |
| | History | 226 (8.0) | 223 (7.7) | 235 (7.7) | 684 |
| | Art | 298 (10.6) | 272 (9.4) | 270 (8.8) | 840 |
| | Music | 150 (5.3) | 161 (5.5) | 203 (6.6) | 514 |
| | Education | 174 (6.2) | 181 (6.2) | 227 (7.4) | 582 |
| | Economics | 276 (9.8) | 265 (9.1) | 318 (10.4) | 859 |
| | Sum | 2,809 (100) | 2,904 (100) | 3,059 (100) | 8772 |

before this research. The names of the college student participants were coded to protect their privacy.

**Instruments.** Testers must undergo rigorous training before the test to ensure safety and accuracy. All tests used the same manufacturer and batch of instruments according to the standard procedures for college students prescribed in the 2021 CNSSCH. Measurement instruments were calibrated everyday morning and at fixed intervals throughout the day (5 times at least). Every physical fitness indicator test was required to be completed at the same fixed time of the day to reduce data deviation caused by different test times. In this study, seven physical fitness indicators were used, which were BMI (calculated by height and weight), forced vital capacity, 50-m dash, standing long jump, sit and reach, upper body muscle strength (pull-ups for boys and one minute sit-ups for girls), and endurance run (800 m for girls and 1000 m for boys).

1. Body mass index (BMI) = weight (kg)/height (cm). Weight and height were simultaneously measured to the nearest 0.1 kg and 0.1 cm with the electronic instrument (TSN200-ST, Tishineng Sports, made in China), respectively. College participants must upright their bodies, look forward, and take off their shoes and hats.

2. Forced vital capacity was measured using the vital capacity meter (TSN200-FH, Tishineng Sports, made in China). Each participant had a new blowing nozzle. During the test, the participants are in a natural standing position, holding the handle of the venturi tube, and trying their best to inhale deeply. Then, they used the mouth piece to slowly exhale until they were unable. The value displayed on the display screen was the vital capacity value. A total of 2 tests were conducted and recorded the maximum value in milliliters without retaining decimals. The interval between the two tests should not exceed 15 seconds.

3. The 50-m dash was measured by the test instrument (TSN200-WP, Tishineng Sports, made in China). The college student participants heard the starting signal and ran as fast as possible to the finish line. 50-m dash was only measured once and the time precision was 0.1s.

4. Standing long jump was measured on the specialized long jump mats (TSN200-TY, Tishineng Sports, made in China), which allowed participants to try twice and record the best score. Each college student stands in front of the starting line with both feet shoulder-width apart, jumping with both feet simultaneously. The longest distance between the starting line and the heel of the nearest shoe was recorded after measuring twice.

5. Sit and reach was measured with the instrument model TSN200-TQ, Tishineng Sports, made in China. The participants removed their shoes, straightened their legs, and sat down with their toes against the testing board. College students slowly bent over and straightened their arms to push the ruler as far as possible, the farthest distance was the best score that needed to be recorded after twice trials.

6. Upper body muscle strength, including pull-ups for boys (TSN200-YT, Tishineng Sports, made in China), and one minute sit-ups for girls (TSN200-YW, Tishineng Sports, made in China). Pull-ups were tested on horizontal bars, using the back hand method. Upper body strength was used to pull the entire body up. Each move required an elbow angle greater than 90 degrees to record. One minute sit-ups recorded the number of sit-ups in 60 seconds. Participants should bend their knees 90 degrees and lie on a soft cushion. Reaching or passing their knees with their elbows is considered a complete sit-up task. Pull-ups and sit-ups were only tested once.

7. Endurance runs including 1000 m for boys and 800 m for girls, were measured using the long distance running instrument (TSN200-CP, Tishineng Sports, made in China). The test

was conducted on a standard track of 400 m, 2.5 laps for boys and 2 laps for girls. Upon hearing the starting signal, participants ran as fast as they could to the finishing line, recording running time accurately to 0.1s.

The test instruments mentioned above are all bluetooth, which could automatically save and upload data in a standard format.

All seven physical fitness indicators mentioned above were converted to Z-scores via SPSS. The physical fitness indicator (PFI) was obtained using these indicators' Z-scores. The Z-scores for the 50-m dash and endurance run (800 m and 1000 m) were reversed, because lower times reflect better performances in these three tests. Therefore, PFI = Z BMI + Z forced vital capacity–Z 50-m dash + Z standing long jump + Z sit and reach +Z upper body muscle strength −Z endurance run.

## Statistical analysis

The physical fitness record was expressed as the mean±standard deviation. The chi-square test was utilized to assess the disparities in total physical fitness scores among college students across different grades. One-way ANOVA and LSD tests were conducted to compare physical fitness scores with different grades in the same major. Independent sample $t$-tests were used to compare the differences in every physical fitness indicator for different majors and sexes.

Pearson's correlations among 12 majors for every grade were conducted to study the significant corrections between every two physical fitness indicators. The BMI and PFI for college students of different grades were investigated using a nonlinear regression model. All statistical analyses mentioned in this study were conducted with SPSS software version 25.0 (SPSS Inc., Chicago, IL, USA).

## Results

Based on 12 different majors and physical fitness record levels, the number of college students in three grades was statistically analyzed in Table 2. Most college students fell near the passing line (75.3%), following students who perform good (23.6%) and fail (0.6%), with the least number of students performing excellent (0.5%). In the excellent record levels, the number of sophomore students (25) exceeded freshmen (13) and juniors (5). Similarly, with good record levels, the number of sophomore students (947) exceeded freshmen (652) and juniors (470). But, in pass and fail record levels, the number of junior students (2,559 and 25, respectively) exceeded freshmen (2,124 and 20, respectively) and sophomores (1,924 and 8, respectively). Major economics had the most (10) in excellent, but majors of math, history, and major art had the lowest (1). Major physics had the most fail (14), while majors of Chinese and education had zero fail students.

An analysis of the variations in total physical fitness scores among college students of different academic years revealed marked discrepancies across all 12 disciplines, with 10 demonstrating highly significant differences, except for the biology and music fields.

Table 3 shows the mean and standard deviation of the total physical fitness scores for the three grades with different majors. It can be observed that sophomores performed better than freshmen except major music, and juniors were the worst for all 12 majors. In freshman year, major music had the highest score (78.01±4.58), and major art had the lowest score (73.85 ±6.02). In sophomore year, major English had the highest score (79.29±5.03), and major art had the lowest score (74.97±5.53). In junior year, major education had the highest score (76.26 ±4.81), and major art had the lowest score (72.59±4.84). The results of the one-way ANOVA analysis and pairwise comparison of LSD showed that there were significant differences ($P<$

**Table 2. Statistics of the number of college students in different record levels and grades based on 12 majors.**

| Subject | Major | Excellent | | | Good | | | Pass | | | Fail | | | X² | P |
|---|---|---|---|---|---|---|---|---|---|---|---|---|---|---|---|
| | | A | B | C | A | B | C | A | B | C | A | B | C | | |
| Sciences | Math | 1 | 0 | 0 | 48 | 72 | 35 | 206 | 206 | 247 | 3 | 0 | 5 | 23.36 | 0.001** |
| | Physics | 1 | 1 | 0 | 42 | 86 | 38 | 204 | 170 | 212 | 2 | 4 | 8 | 35.11 | 0** |
| | Chemistry | 0 | 2 | 2 | 40 | 99 | 51 | 158 | 163 | 183 | 3 | 0 | 3 | 28.45 | 0** |
| | Biology | 0 | 6 | 0 | 67 | 84 | 76 | 168 | 184 | 204 | 3 | 1 | 4 | 14.37 | 0.026* |
| Humanities | Chinese | 0 | 2 | 1 | 50 | 71 | 38 | 185 | 173 | 221 | 0 | 0 | 0 | 17.43 | 0.002** |
| | English | 0 | 4 | 0 | 69 | 113 | 34 | 193 | 138 | 214 | 0 | 0 | 1 | 70.87 | 0** |
| | Law | 2 | 1 | 0 | 58 | 92 | 22 | 179 | 130 | 207 | 3 | 0 | 0 | 68.89 | 0** |
| | History | 1 | 0 | 0 | 54 | 78 | 36 | 170 | 145 | 199 | 1 | 0 | 0 | 28.01 | 0** |
| | Art | 0 | 0 | 1 | 45 | 45 | 15 | 249 | 225 | 251 | 4 | 2 | 3 | 20.25 | 0.003** |
| | Music | 1 | 3 | 1 | 50 | 41 | 37 | 99 | 117 | 164 | 0 | 0 | 1 | 14.01 | 0.03* |
| | Education | 3 | 0 | 0 | 49 | 75 | 47 | 122 | 106 | 180 | 0 | 0 | 0 | 28.07 | 0** |
| | Economics | 4 | 6 | 0 | 80 | 91 | 41 | 191 | 167 | 277 | 1 | 1 | 0 | 50.39 | 0** |
| Sum | | 13 | 25 | 5 | 652 | 947 | 470 | 2,124 | 1,924 | 2,559 | 20 | 8 | 25 | | |
| | | 43 (0.5%) | | | 2,069 (23.6%) | | | 6,607 (75.3%) | | | 53 (0.6%) | | | | |

Note: A-Freshman, B-Sophomore, C-Junior

*indicates significant difference

**indicates extremely significant difference.

0.01) in the total physical fitness scores across all 12 majors in the three grades. Compare with sophomores, freshmen scores showed that the differences in every major were significant ($P < 0.05$ or $P < 0.01$) except for majors of biology and economics. Compared with juniors, freshmen showed that the differences in every major were significant ($P < 0.01$) except for majors of math, chemistry, Chinese, and biology. There were significant differences ($P < 0.01$) in physical fitness scores across all 12 majors between sophomores and juniors.

**Table 3. Comparison of total physical fitness test scores with different grades in every major.**

| Major | Freshman (A) | Sophomore (B) | Junior (C) | F/P | Difference value/P | | | | | |
|---|---|---|---|---|---|---|---|---|---|---|
| | Mean±SD | Mean±SD | Mean±SD | | A-B | P | A-C | P | B-C | P |
| Math | 74.62±6.01 | 76.49±4.95 | 74.03±5.81 | 14.71/0.000 | -1.87 | 0.000 | 0.59 | 0.22 | 2.46 | 0.000 |
| Physics | 74.44±5.98 | 76.17±6.58 | 72.88±6.43 | 17.47/0.000 | -1.72 | 0.002 | 1.57 | 0.006 | 3.29 | 0.000 |
| Chemistry | 74.93±5.89 | 78.36±5.67 | 74.45±6.21 | 23.71/0.000 | -3.43 | 0.000 | -0.52 | 0.36 | 2.91 | 0.000 |
| Chinese | 76.01±5.08 | 77.43±5.04 | 75.58±4.59 | 9.69/0.000 | -1.41 | 0.002 | 0.44 | 0.32 | 1.85 | 0.000 |
| English | 77.04±4.59 | 79.29±5.03 | 75.18±4.97 | 48.59/0.000 | -2.26 | 0.000 | 1.85 | 0.000 | 4.11 | 0.000 |
| Law | 76.14±5.96 | 78.22±5.21 | 73.98±5.64 | 35.13/0.000 | -2.08 | 0.000 | 2.16 | 0.000 | 4.24 | 0.000 |
| History | 76.15±5.47 | 77.79±5.09 | 74.52±5.58 | 21.07/0.000 | -1.64 | 0.001 | 1.63 | 0.001 | 3.27 | 0.000 |
| Art | 73.85±6.02 | 74.97±5.53 | 72.59±4.84 | 11.75/0.000 | -1.12 | 0.016 | 1.26 | 0.007 | 2.38 | 0.000 |
| Music | 78.01±4.58 | 76.79±5.51 | 75.37±5.72 | 10.72/0.000 | 1.21 | 0.046 | 2.64 | 0.000 | 1.43 | 0.012 |
| Education | 77.78±4.85 | 78.93±4.39 | 76.26±4.81 | 16.73/0.000 | -1.15 | 0.021 | 1.52 | 0.001 | 2.68 | 0.000 |
| Biology | 76.94±5.49 | 77.37±5.73 | 75.99±5.78 | 4.28/0.014 | -0.43 | 0.390 | 0.94 | 0.059 | 1.37 | 0.004 |
| Economics | 77.46±5.35 | 78.03±5.98 | 74.59±5.57 | 34.33/0.000 | -0.57 | 0.220 | 2.82 | 0.000 | 3.39 | 0.000 |

Further comparisons of the individual physical fitness test scores in different grades were carried out for every major (S1 Table). For all 12 majors, readers can find the individual physical fitness test scores for all seven physical fitness indicators from freshmen to sophomores to juniors and the significant differences in $P$ values between every two grades. It was worth noting that there were significant differences in the pairwise comparison of the three grades for majors of law, art, biology, music, and economics in 50-m dash, major education in standing long jump, and major biology in endurance run.

**Table 4. Comparison of individual and total physical fitness test scores between Humanities and Sciences of the same sex.**

| Sex | Physical fitnessindicators | Humanities | Sciences | T-test | |
|---|---|---|---|---|---|
| | | Mean±SD | Mean±SD | F | P |
| Boys | BMI | 14.27±1.29 | 14.34±1.24 | 8.73 | 0.003 |
| | forced vital capacity | 11.87±0.98 | 11.72±1.95 | 1.31 | 0.25 |
| | 50-m dash | 15.42±1.82 | 15.66±1.89 | 3.75 | 0.053 |
| | standing long jump | 6.85±0.95 | 6.91±0.88 | 6.91 | 0.009 |
| | sit and reach | 7.67±1.07 | 7.62±1.05 | 0.73 | 0.39 |
| | upper body muscle strength | 2.39±2.91 | 2.51±2.89 | 1.81 | 0.18 |
| | endurance run | 13.35±1.95 | 13.48±2.05 | 0.061 | 0.81 |
| | Total score | 71.99±6.37 | 72.25±6.53 | 0.16 | 0.69 |
| Girls | BMI | 14.59±1.03 | 14.69±0.92 | 52.69 | 0 |
| | forced vital capacity | 11.74±1.94 | 11.65±1.78 | 38.06 | 0 |
| | 50-m dash | 14.19±1.41 | 14.24±1.28 | 6.22 | 0.013 |
| | standing long jump | 7.28±0.96 | 7.48±0.95 | 0.99 | 0.32 |
| | sit and reach | 8.30±1.12 | 8.36±0.14 | 1.45 | 0.23 |
| | upper body muscle strength | 6.82±0.81 | 6.93±0.87 | 9.31 | 0.002 |
| | endurance run | 14.36±1.56 | 14.65±1.51 | 2.62 | 0.11 |
| | Total score | 77.27±4.61 | 77.99±4.39 | 7.21 | 0.007 |

To compare the difference between the humanities and sciences, independent sample *t*-tests were used in seven physical fitness indicators for different gender in Table 4. The total score showed that girl students had higher scores (77.27±4.61 and 77.99±4.39, respectively) than boy students (71.99±6.37 and 72.25±6.53, respectively) in both humanities and sciences majors. For boy students, major sciences performed better in BMI, 50-m dash, standing long jump, upper body muscle strength, and endurance run than major humanities. For girl students, major science performed better in other six physical fitness indicators, except for forced vital capacity, than major humanities. There were significant differences ($P < 0.01$) between major humanities and sciences in BMI and standing long jump for boys, and in BMI, forced vital capacity, 50-m dash, and upper body muscle strength for girls.

Table 5 shows the results of Pearson's correlations between every two physical fitness indicators in all 12 majors for every grade divided by gender. There were no significant correlations between BMI and forced vital capacity and between forced vital capacity and upper body muscle strength in all three grades for boys. For boy students, there were also no significant correlations between BMI and sit and reach in freshman year; BMI and sit and reach, 50-m dash and sit and reach in sophomore year; and BMI and upper body muscle strength, forced vital capacity and 50-m dash, and forced vital capacity and endurance run in junior year. For girls, there were no significant correlations between BMI and forced vital capacity, BMI and standing long jump, forced vital capacity and 50-m dash, forced vital capacity and upper body muscle strength, forced vital capacity and endurance run, and 50-m dash and sit and reach in freshman year; BMI and forced vital capacity, BMI and upper body muscle strength, forced vital capacity and endurance run, and 50-m dash and sit and reach in sophomore year; and BMI and standing long jump in junior year. Besides the correlations mentioned above, there were statistically significant correlations ($P < 0.05$ or $P < 0.01$) between every two physical fitness indicators used in this study.

The relationship between BMI Z-scores and PFI Z-scores for girls (Fig 1) and boys (Fig 2) college student were estimated using the nonlinear regression model (SPSS: analysis-regression-curve estimation-quadratic model). All figures revealed the inverted U-shaped curves in

**Table 5. Correlations between physical fitness indicators for college students in every grade by sex (r-value).**

| Sex | Grades | A/B | A/C | A/D | A/E | A/F | A/G | B/C | B/D | B/E | B/F | B/G | C/D | C/E | C/F | C/G | D/E | D/F | D/G | E/F | E/G | F/G |
|---|---|---|---|---|---|---|---|---|---|---|---|---|---|---|---|---|---|---|---|---|---|---|
| Boys | Freshman | 0.28 | 0.12[a] | 0.13[a] | 0.056 | 0.16[a] | 0.18[a] | 0.17[a] | 0.083[b] | 0.21[a] | -0.024 | 0.11[a] | 0.19[a] | 0.12[a] | 0.15[a] | 0.34[a] | 0.13[a] | 0.22[a] | 0.31[a] | 0.12[a] | 0.13[a] | 0.28[a] |
|  | Sophomore | -0.003 | 0.12[a] | 0.16[a] | -0.014 | 0.1[a] | 0.21[a] | 0.16[a] | 0.095[a] | 0.029 | -0.06 | 0.11[a] | 0.4[a] | -0.051 | 0.25[a] | 0.3[a] | 0.075[b] | 0.31[a] | 0.32[a] | 0.031 | 0.052 | 0.28[a] |
|  | Junior | -0.004 | 0.16[a] | 0.15[a] | 0.12[a] | 0.047 | 0.14[a] | 0.069 | 0.1[a] | 0.071[b] | 0.069 | 0.037 | 0.41[a] | 0.12[a] | 0.24[a] | 0.31[a] | 0.17[a] | 0.26[a] | 0.23[a] | 0.13[a] | 0.13[a] | 0.19[a] |
| Girls | Freshman | 0.018 | 0.07[a] | 0.043 | 0.06[a] | 0.068[a] | 0.11[a] | 0.003 | 0.11[a] | 0.079[a] | 0.024 | 0.033 | 0.43[a] | 0.031 | 0.18[a] | 0.36[a] | 0.16[a] | 0.15[a] | 0.34[a] | 0.14[a] | 0.074[a] | 0.17[a] |
|  | Sophomore | 0.001 | 0.048[b] | 0.052[b] | 0.078[a] | 0.039 | 0.088[a] | 0.05[b] | 0.11[a] | 0.04 | 0.071[a] | 0.018 | 0.45[a] | 0.03 | 0.21[a] | 0.39[a] | 0.14[a] | 0.23[a] | 0.32[a] | 0.069[a] | 0.098[a] | 0.22[a] |
|  | Junior | 0.045[b] | 0.051[b] | 0.04 | 0.051[b] | 0.066[a] | 0.071[a] | 0.14[a] | 0.089[a] | 0.044[b] | 0.045[b] | 0.064[a] | 0.44[a] | 0.12[a] | 0.26[a] | 0.37[a] | 0.18[a] | 0.28[a] | 0.32[a] | 0.14[a] | 0.14[a] | 0.27[a] |

Note

[a] < 0.01

[b] < 0.05.

A, BMI; B, forced vital capacity; C, 50- m dash; D, standing long jump; E, sit and reach; F, upper body muscle strength; G, endurance run.

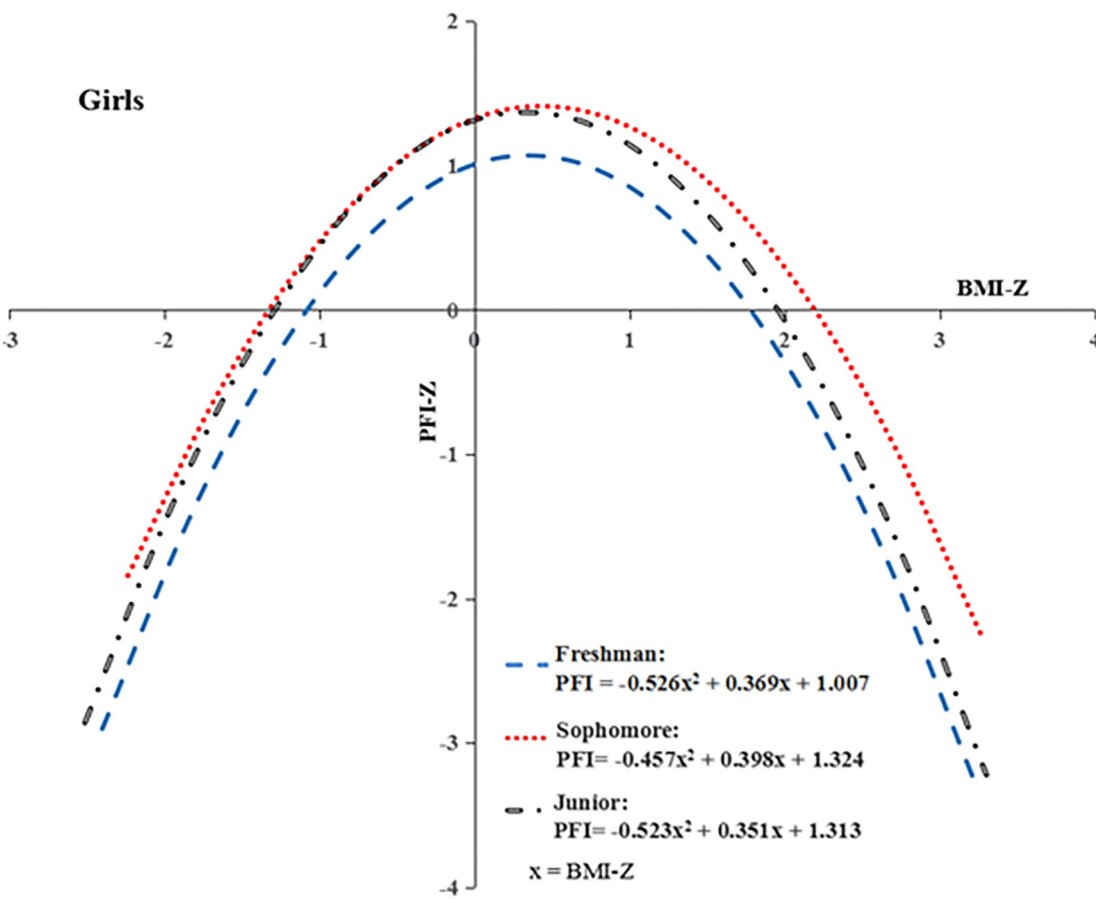

**Fig 1. Linear relationship between BMI Z-scores and PFI Z-scores for girls college students in each grade.**

three different grades for both girls and boys. The results of the nonlinear regression model indicated that higher and lower BMI negatively affected PFI for both boys and girls college students. It found that, overall, the girl college students had notably higher PFI Z-scores than boys with the same BMI Z-scores. Freshman students performed the worst. Sophomore students performed better than junior students for girls, and the junior students slightly outperformed sophomores, with BMI Z-scores ranging from -0.5 to 1.7.

## Discussion

Based on the physical fitness record levels, the proportion of excellent (0.5%) and fail (0.6%) groups were the least. The physical fitness score of most college students fell near the passing line (75.3%) [26], following students who performed good (23.6%). This suggests that the physical fitness of college students nowadays is poor [16, 22, 27, 28]. Upon investigating potential reasons, the overwhelming majority of college students were highly impacted by screen time, obesity, and sedentary, leading to less exercise time [29]. This phenomenon grew even worse during the COVID-19 lockdown period for college students [30]. Kidokoro et al. found that the population health declined for Japanese children and adolescents during the COVID-19 pandemic [31]. This fully proves the necessity and significance of China's national strategy of public fitness. Moreover, this study found that sophomore students performed better than

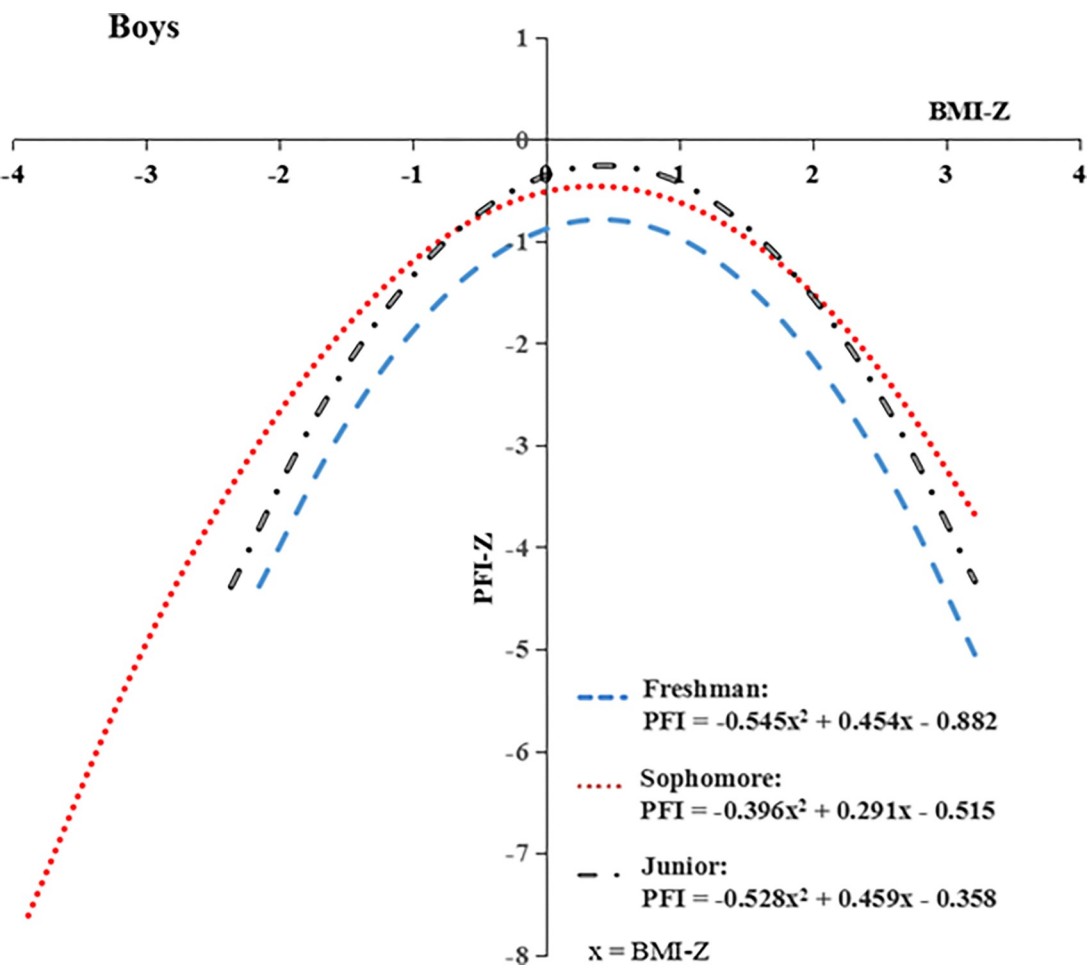

**Fig 2. Linear relationship between BMI Z-scores and PFI Z-scores for boys college students in each grade.**

freshman and junior year. This finding was consistent with Cao H., who revealed that sophomores had the best pass rate, following freshmen, juniors and seniors [32].

Generally speaking, sophomores had better total physical fitness scores than freshmen, and juniors had the lowest values, for all the eleven majors except for music. This trend was in line with the current situation of universities revealed by Song Y. [33] and Hu et al. [34], where the physical fitness of senior students was poorer than that of lower grade students [26]. On the one hand, most colleges do not offer specialized physical education classes for students in their third and fourth years, no longer ideologically valuing physical sports [35]. On the other hand, senior students feel pressured to graduate or find a job, they do not have time to exercise [36]. It is crucial to cultivate a habit of exercise among college students, which will benefit them for their lifetime [37]. Thus, freshman and sophomore students must proficiently master one or two sportskills [38, 39], be able to play games, and find joy in playing sports, which is conducive to developing their sport habits. Constructing a healthy China requires the support of people's good exercise habits [40, 41], especially for college students who are valuable talent and the future of China.

For freshman students, major music had the highest score (78.01±4.58), and major art had the lowest score (73.85±6.02). In sophomore year, major English had the highest score (79.29 ±5.03), and major art had the lowest score (74.97±5.53). Wang Z. conducted a study on the

physical health of college students of 11 majors in Lvliang College, but had no total score to compare with this study [42]. In junior year, major education had the highest score (76.26 ±4.81), and major art had the lowest score (72.59±4.84). Both majors of art and physics had the worst performance in all three grades in the humanities and sciences, respectively, which urgently require sufficient attention and further investigation. Hu et al. found that there were significant differences among different majors, with major agronomy (belonging to the sciences) having the best performance and Chinese studies (belonging to the humanities) performing the worst [34]. Readers can find the ranking of 17 majors in his original text [34].

There were significant differences ($P < 0.01$) in the total physical fitness scores across the three grades in all 12 majors [16] and most results of pairwise comparisons from freshmen to juniors, which means each college student's lifestyle and exercise habits are all different [42–45]. It was worth mentioning that there were significant differences ($P < 0.01$) in the physical fitness scores across all majors between sophomores and juniors [33, 46]. This conclusion was similar to Song Y., who revealed that seniors had the worst scores [33]. Sophomore students had lower values than other students in all 12 majors, which was an important discovery of this article [46]. Shen et al. compared the daily physical activity levels between professional female graduates and non-professional, the different majors being one of the important reasons [43]. Sheng et al. found that most students do not engage in other heavy physical activities after completing physical education classes [47]. Senior students no longer have mandatory physical education classes, resulting in a significant decrease in their chances of engaging in sports.

Pairwise comparisons revealed more significant differences in individual physical fitness indicators among the three grades in the humanities (majors of law, art, music, education, and economics) than in sciences (major biology). This suggests great imbalances among individual physical fitness indicators in the humanities, while the physical fitness development of science students is relatively balanced. These results conform to many people's preconception that major sciences have more physical activity than humanities students'. Ping Y. found no significant differences in academic achievements between humanities and science students majoring in social sports [48], which means that the physical fitness development of sports majors in the humanities and sciences is relatively balanced.

There are significant differences ($P < 0.01$) in BMI between humanities and sciences students for both boys and girls, with major humanities having lower BMI values than major sciences. Zhang et al. found that there was significant difference among the humanities, sciences, and arts ($P < 0.001$) [49]. In the standing long jump for boys, science students scored significantly ($P < 0.01$) higher than humanities students [49]. Due to the different majors, humanities students may have fewer physical activities and longer sedentary times than science students during non-physical education classes [50]. There are also significant differences ($P < 0.01$) in forced vital capacity, 50-m dash, and upper body muscle strength between major humanities and science for only girls. This phenomenon indicates that the physical fitness of girls is more influenced by different majors. This may be related to the difference between liberal humanities and science courses [49]. There are more experimental courses in science and far more opportunities for exercise, while there are more liberal arts and history courses in humanities with longer sedentary hours [50].

Pearson's correlations among seven physical fitness indicators in three grades ranged from 0.001 to 0.45. Most r-values showed significant correlations ($P < 0.05$ or $P < 0.01$), suggesting a close relationship existed with each other. There were no significant correlations between many physical fitness indicators for boy juniors (Table 5). This maybe the reason for junior students performing worse than freshmen and sophomores in most majors [33, 46] (see Table 3).

The results of this research revealed that PFI had a parabolic trend with BMI for both boy and girl college students in Jiangxi province. This finding was similar to Bi et al., who found inverted U-shaped curves between BMI Z-scores and PFI Z-scores for the young aged 7–18 in Xinjiang from 1985 to 2014 [51]. The results indicated that higher and lower BMI negatively affected the PFI for both boys and girls college student [52]. The normal weights and heights can help attain a better PFI. To reach this goal, Jiang et al. suggested universities help students develop proper exercise habits and enhance the ideological awareness of the importance of sports [53]. Girl college students had much higher PFI Z-scores than boys with the same BMI Z-score in this research. This trend consistent with the findings of Bi et al. [51, 54].

Freshman students performed the worst (see Figs 1 and 2). This maybe related to the just completed college entrance examination and the almost three-month-long summer vacation. Dai C. believed that freshmen have not developed good lifestyle habits and daily physical exercise habits, mostly focused on the online world for daily attention [55]. Sophomore students performed better than junior students for girls, but junior students slightly outperformed sophomores, with BMI Z-scores ranging from -0.5 to 1.7. Most colleges in China do not have physical education (PE) classes for junior and senior students. Combined with senior internships and job hunting, this exacerbates the lack of time for college students to exercise. The life and behavioral habits of college students, excessive screen time, and master's entrance examination pressure have contributed to the sharp increase in sedentary time [29, 44, 46, 56], which might be another reason for the poor physical fitness of advanced college students. Therefore, to improve the physical fitness levels of college students, especially advanced college students, necessary measures must be carried out to control excessive thinness or obesity, with treatment strategies implemented and targeted for college students of different grades and majors.

## Limitations of the study

This study offers a comparative examination of the physical health status of college students in Jiangxi Province through two lenses: grades and majors. However, several constraints must be acknowledged. Firstly, the existing research on Chinese college students' physical health is limited, complicating comparative analyses. Secondly, the sample size employed in this study is modest, potentially compromising its representativeness of Jiangxi Province's college students' physical health status. To overcome these limitations, future inquiries should engage more scholars to conduct thorough investigations at local provincial and municipal universities, as well as foster multi-school, inter-provincial, and inter-municipal collaborations. Such an approach will facilitate a more holistic understanding and precise evaluation of the physical health status of college students in Jiangxi Province.

## Conclusions

The physical fitness of college students is poor today because most scores were near the passing line (75.3%). Sophomore students had lower scores than freshman and junior students across all 12 majors. From freshman to junior year, majors of music (78.01±4.58), English (79.29 ±5.03), and education (76.26±4.81) had the highest scores, respectively, but major art consistently scored the lowest, which were 73.85±6.02, 74.97±5.53, and 72.59±4.84, respectively. Different college grades and majors play important roles in the physical fitness of students in Jiangxi province. This study found that, overall, senior and humanities students had much poorer physical fitness. The abnormal BMI negatively affected the PFI for both boy and girl college students. Issuingan exercise prescription is an effective way to solve this problem, namely, setting up different sports programs based on the characteristics of college students of

different grades and majors so that the intensity of exercise meets the requirements of the course objectives. Further research on these exercise prescriptions is urgently needed for both whole classes and individual students.

## Supporting information

**S1 Table.**
(XLS)

**S1 Dataset.**
(XLSX)

## Author Contributions

**Conceptualization:** Jianzhong Sun, Cunjian Bi.

**Data curation:** Chan Lin, Lei Wang.

**Investigation:** Chan Lin, Lei Wang.

**Methodology:** Jianzhong Sun, Bin Qiao.

**Software:** Jianzhong Sun, Bin Qiao.

**Writing – original draft:** Cunjian Bi.

**Writing – review & editing:** Jianzhong Sun, Bin Qiao.

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
