## [Decision Letter · Decision Letter 0]

24 Apr 2024

PONE-D-23-32339Comparative study on physical fitness of different grades and majors for college students of Jiangxi provincePLOS ONE

Dear Dr. Sun,

Thank you for submitting your manuscript to PLOS ONE. After careful consideration, we feel that it has merit but does not fully meet PLOS ONE’s publication criteria as it currently stands. Therefore, we invite you to submit a revised version of the manuscript that addresses the points raised during the review process.

You have previously received a decision from the Academic editor via email, this is a formal confirmation of that decision. I am aware that you have already conducted your revisions so please follow the below instructions on how to submit your revised manuscript.

We look forward to receiving your revised manuscript.

Kind regards,

Emma Campbell, Ph.D

Staff Editor

PLOS ONE

On behalf of: 

Lütfullah Türkmen

Academic Editor 

PLOS ONE

Journal Requirements:

"This work was supported by “School level scientific research project of Chizhou University in 2022 (Award No.CZ2022YJRC03)”, “Anhui province 2022 provincial quality engineering project for higher education institutions (Award No. 2022jxms116)” and “Chizhou university campus research center, 2021, Sports and health promotion center”."    

4. In this instance it seems there may be acceptable restrictions in place that prevent the public sharing of your minimal data. However, in line with our goal of ensuring long-term data availability to all interested researchers, PLOS’ Data Policy states that authors cannot be the sole named individuals responsible for ensuring data access (http://journals.plos.org/plosone/s/data-availability#loc-acceptable-data-sharing-methods).

Additional Editor Comments:

Dear Author, after your final revision and the evaluation of your paper, I am going to make a decision related to your paper further toward accepting for the publication. However, could you submit the final version of your paper to the PLOS system that covering your latest revisions based on the reviewer's comments.

Reviewers' comments:

Reviewer's Responses to Questions

**Comments to the Author**

1. Is the manuscript technically sound, and do the data support the conclusions?

Reviewer #1: Yes

2. Has the statistical analysis been performed appropriately and rigorously? 

Reviewer #1: Yes

3. Have the authors made all data underlying the findings in their manuscript fully available?

Reviewer #1: Yes

4. Is the manuscript presented in an intelligible fashion and written in standard English?

Reviewer #1: No

5. Review Comments to the Author

Reviewer #1: Abstract

Please add the participants’ characteristics (gender, age, grade ……etc.).

Please indicate the seven physical fitness indicators.

Background

Most Swedish ....... Please indicate worldwide statistics and then give some examples.

Poor physical condition not only affects the immune system and increases risk factors for death. Please explain further the harmful effects of poor physical condition.

Latest data .......... year. Need reference(s).

Poor lifestyle habits. Please explain.

Page 3, line 2 - The authors mention two levels of significance for a negative correlation! Moreover, this correlation is well demonstrated and makes sense.

General comments:

The problematic is unclear.

The authors should add the study's hypotheses.

Despite the extensive research on physical fitness, the authors did not present the introduction section in a clear and organized way. They have merely presented fragmented, unconnected data. I kindly request the authors to replicate this part by precisely delineating the concept of fitness, its constituent elements, its impact on health, and the various factors that influence it. Subsequently, they can provide an assessment about the global and national state of physical fitness. In addition, it is imperative to elucidate the practical value of a high degree of physical fitness within the target group under investigation, as well as the underlying rationale for conducting this study.

Methods

CNSSCH? Please explain or add a reference.

Please add the sampling technique used and explain all the sampling steps.

Please add participant characteristics (gender, age, grade, ......).

Was the tester's accuracy checked before testing began?

In the Statistical analysis section, the authors state that the BMI and PFI of students in different classes were studied using a non-linear regression model. Please justify and explain this choice.

Results

Physical fitness levels should be described in the "Methods" section.

In my opinion, comparing frequencies using chi-squares will bring more precision to the results in Table 2.

In Table 3, p-value ≠ 0, the authors might use 0.000 or <0.001.

In the tables, please add in brackets the number of participants by grade and major.

In Table 4, for each variable, the t-test has been run twice, which may generate a Type I error.

The figures are the study's best aspect, however the authors have supplied little information on how they were created, particularly how the second-degree equations were generated from a non-linear regression.

Discussion

The authors are invited to compare their findings with those of other Chinese regions in the opening paragraph. They might also discuss Jiangxi province's unique characteristics, which could account for the observed variations.

Lines 5 to 7 on page 15: Are these the primary findings of this study or are they the findings from references 25 and 26 provided here? Authors are advised to prioritize the presentation of their primary findings, followed by a comparative analysis with the results of other studies.

The same applies to references 36 and 37 on line 20.

Page 16, line 4: [10]?

Page 17, lines 13-21: The authors have restated the correlation findings in this section. I urge them to only present and discuss the primary relationships that have been observed.

Please add the study's limitations.

General Comment: This study looked at BMI and PFI for three grade levels with age gaps of two to three years or more. The impact of growth on these issues, however, was never mentioned in the discussion section.

Conclusions

The main results of the study should be mentioned here.

References

Ref.8: The citation is inaccurate.

Ref. 11: I see no connection with the present study. Please correct.

Ref.13: The citation is inaccurate.

Ref.26: The citation is inaccurate.

Ref.41: The citation is inaccurate.

General Comment: Several references were either absent from online sources or were published exclusively in local journals. I strongly encourage authors to thoroughly examine all references, accurately document them, and exclusively utilize those from reputable academic journals.

6. PLOS authors have the option to publish the peer review history of their article (what does this mean?). If published, this will include your full peer review and any attached files.

Reviewer #1: **Yes: **Dr. Mohamed Ahmed Said

---

## [Author Response · Author response to Decision Letter 0]

26 Apr 2024

PONE-D-23-32339

Comparative study on physical fitness of different grades and majors for college students of Jiangxi province

PLOS ONE

Abstract

 Please add the participants’ characteristics (gender, age, grade ……etc.). 

Response: Added it in abstract (lines 7-8, page 1, in Revised version), 2,404 boys and 6,368 girls of Chinese college students in Jiangxi province, from freshman to junior years, aged 17-22.

 Please indicate the seven physical fitness indicators. 

Response: Added it in abstract (lines 10-11, page 1, in Revised version), body mass index (BMI), forced vital capacity, 50-m dash, standing long jump, sit and reach, upper body muscle strength, endurance run.

Background

 Most Swedish ....... Please indicate worldwide statistics and then give some examples. 

Response: Added it in background (the 1-4 to last line, page 2, in Revised version), “In accordance with the latest World Health Organization (WHO) guidelines for physical activity, adults should aim to engage in at least 150-300 minutes of moderate intensity aerobic activity weekly, or 75-150 minutes of high-intensity aerobic activity weekly, or a combination of both moderate and high-intensity activities.”

 Poor physical condition not only affects the immune system and increases risk factors for death. Please explain further the harmful effects of poor physical condition.

Response: Added it in background (line 4-7, page 3, in Revised version), “Physical inactivity is a significant contributor to cardiovascular disease, diabetes, and obesity, which can heighten the risk of developing conditions such as hypertension, hyperlipidemia, osteoporosis, depression, diabetes, obesity, and heart disease, ultimately leading to life-threatening outcomes.”

 Latest data .......... year. Need reference(s).

Response: Added the following reference (Ref. 9, lines 10-12, page 23, in Revised version), Ho C C, Lee P F, Chen H L, et al. Poor health-related physical fitness performance increases the overweight and obesity risk in elderly people from Taiwan. 2020. DOI:10.21203/rs.3.rs-45860/v2.

 Poor lifestyle habits. Please explain. 

Response: (lines 12-13, page 3, in Revised version) such as physical inactivity and poor diets, added this in sentence.

 Page 3, line 2 - The authors mention two levels of significance for a negative correlation! Moreover, this correlation is well demonstrated and makes sense.

Response: Thanks.

General comments:

 The problematic is unclear. 

Response: Scientific questions added in abstract (line 1-2, page 1, in Revised version), “Research to date has not provided a clear understanding of how different grades and majors affect the physical fitness of college students”.

 The authors should add the study's hypotheses. 

Response: Study’s hypotheses added in abstract (lines 3-4, page 1, in Revised version), “It is postulated that there are significant disparities in physical health among college students of different grades and majors.”

 Despite the extensive research on physical fitness, the authors did not present the introduction section in a clear and organized way. They have merely presented fragmented, unconnected data. I kindly request the authors to replicate this part by precisely delineating the concept of fitness, its constituent elements, its impact on health, and the various factors that influence it. Subsequently, they can provide an assessment about the global and national state of physical fitness. In addition, it is imperative to elucidate the practical value of a high degree of physical fitness within the target group under investigation, as well as the underlying rationale for conducting this study. 

Response: Added the following paragraph in background (lines 16-22, page 3, in Revised version), “Fitness refers to a series of physical exercises designed to enhance muscle development, improve physical strength, enhance body shape, and improve mental well-being through the use of either bare hands or various equipment. It encompasses five key elements: clear fitness goals, a blend of aerobic and anaerobic activities, regular exercise, a balanced diet, and rest and recovery. According to the inaugural Global Health Index, which evaluated 146 countries and regions, the average score was 46.96. China's score ranges between 68.1, slightly above the global average.”

Added the following paragraph in background (lines 7-11, page 4, in Revised version), “To the best of our knowledge, there is currently no comprehensive research on the physical well-being of college students in Jiangxi Province, China. This study aims to serve as a valuable reference for enhancing the physical fitness of college students in Jiangxi province, China.”

Methods

 CNSSCH? Please explain or add a reference. 

Response: Added the following reference (Ref. 25, lines 29-32, page 24, in Revised version), Yuan J M, Sun F, Zhao X, et al. The relationship between mindfulness and mental health among Chinese college students during the closed-loop management of the COVID-19 pandemic: A moderated mediation model. Journal of Affective Disorders, 2023, 327: 137-144. DOI:10.1016/j.jad.2023.02.012.

 Please add the sampling technique used and explain all the sampling steps. 

Response: Revised to the following paragraph (line 25, page 4, in Revised version), “Based on a multistage stratified random cluster sampling method, choose three grades from freshman to junior, including 12 majors (4 science majors and 8 humanities majors) from 4 colleges in Jiangxi province.”

 Please add participant characteristics (gender, age, grade, ......).

Response: Added it in abstract (lines 7-8, page 1, in Revised version), 2,404 boys and 6,368 girls of Chinese college students in Jiangxi province, from freshman to junior years, aged 17-22.

 Was the tester's accuracy checked before testing began? 

Response: Yes, measurement instruments were calibrated everyday morning and at fixed intervals throughout the day (5 times at least), which mentioned in the “Instruments” section (the 7 to last line, page 5, in Revised version).

 In the Statistical analysis section, the authors state that the BMI and PFI of students in different classes were studied using a non-linear regression model. Please justify and explain this choice. 

Response: Figure 1 for girls and Figure 2 for boys. PFI had a parabolic trend with BMI, which was similar to Bi et al., 2019, aim to proving the relationship between BMI and PFI. Detailed information can found in the following reference: Bi CJ, Yang JM, Sun J, Yi S, Wu XY, Zhang F. Benefits of normal body mass index on physical fitness A cross-sectional study among children and adolescents in Xinjiang Uyghur Autonomous Region, China. PLOS ONE.2019; 14(8):1-12.

Results

 Physical fitness levels should be described in the "Methods" section. 

Response: Thanks, moved the following section in the "Methods" section, “All seven physical fitness indicators mentioned above were converted to Z-scores via SPSS. The physical fitness indicator (PFI) was obtained using these indicators’ Z-scores. The Z-scores for the 50-m dash and endurance run (800 m and 1000 m) were reversed, because lower times reflect better performances in these three tests. Therefore, PFI = Z BMI + Z forced vital capacity –Z 50-m dash + Z standing long jump + Z sit and reach +Z upper body muscle strength −Z endurance run. This paragraph moved to the end of the "Methods" section.” (lines 16-21, page 7, in Revised version).

 In my opinion, comparing frequencies using chi-squares will bring more precision to the results in Table 2. 

Response: Added the following paragraph in “Statistical analysis” section, “The chi-square test was utilized to assess the disparities in total physical fitness scores among college students across different grades” (the 4-6 to last line, page 7, in Revised version). And added the X2 and P values in Table 2. 

 In Table 3, p-value ≠ 0, the authors might use 0.000 or <0.001.

Response: All changed in Table 3 according to this revise suggestion (Table 3, page 11, in Revised version).

 In the tables, please add in brackets the number of participants by grade and major. Response: Table 1 includes the number of participants by grade and major (Table 1, page 5, in Revised version).

 In Table 4, for each variable, the t-test has been run twice, which may generate a Type I error. 

Response: Every physical fitness indication, t-test has been run only once divided by gender.

 The figures are the study's best aspect, however the authors have supplied little information on how they were created, particularly how the second-degree equations were generated from a non-linear regression.

Response: Using the nonlinear regression model (SPSS: analysis-regression-curve estimation-quadratic model), and added it in this study (line 3, page 15, in Revised version).

Discussion

 The authors are invited to compare their findings with those of other Chinese regions in the opening paragraph. They might also discuss Jiangxi province's unique characteristics, which could account for the observed variations.

Response: In the "Discussion" section, we compare our findings with those of other Chinese regions (Cao et al.; Song et al.; Hu et al., etc.) since few studies have focused on the physical health of college students. To gain a more accurate understanding and mastery, future research should involve more scholars conducting in-depth studies on local provincial and municipal universities, as well as multi-school, inter-provincial, and inter-municipal cooperation research. These points are further discussed in the "limitations" section (lines 3-9, page 17, in Revised version).

 Lines 5 to 7 on page 15: Are these the primary findings of this study or are they the findings from references 25 and 26 provided here? Authors are advised to prioritize the presentation of their primary findings, followed by a comparative analysis with the results of other studies.

Response: Deleted references of 25 and 26.

 The same applies to references 36 and 37 on line 20. 

Response: Deleted references of 36 and 37.

 Page 16, line 4: [10]? 

Response: Revised.

 Page 17, lines 13-21: The authors have restated the correlation findings in this section. I urge them to only present and discuss the primary relationships that have been observed. 

Response: Deleted the following section, “On the other hand, this proves the scientific nature of China's physical testing programs. If one wants to achieve good physical testing results, perform excellently, and develop in a balanced manner, then all aspects are needed46, 47.”

 Please add the study's limitations. 

Response: Added limitations, “This study offers a comparative examination of the physical health status of college students in Jiangxi Province through two lenses: grades and majors. However, several constraints must be acknowledged. Firstly, the existing research on Chinese college students' physical health is limited, complicating comparative analyses. Secondly, the sample size employed in this study is modest, potentially compromising its representativeness of Jiangxi Province's college students' physical health status. To overcome these limitations, future inquiries should engage more scholars to conduct thorough investigations at local provincial and municipal universities, as well as foster multi-school, inter-provincial, and inter-municipal collaborations. Such an approach will facilitate a more holistic understanding and precise evaluation of the physical health status of college students in Jiangxi Province.” (lines 18-28, page 20, in Revised version).

General Comment: This study looked at BMI and PFI for three grade levels with age gaps of two to three years or more. The impact of growth on these issues, however, was never mentioned in the discussion section. 

Response: The physical fitness of senior and humanities major college students is much weaker and needs sufficient attention, which mentioned in the “Conclusion” section (lines 9-10, page 21, in Revised version).

Conclusions

 The main results of the study should be mentioned here. 

Response: Added the main results in the “Conclusion” section, “The physical fitness of college students is poor today because most scores are near the passing line (75.3%). Sophomore-year students had lower scores than sophomore-year students across all 12 majors. From freshman to junior year, majors of music (78.01±4.58), English (79.29±5.03), and education (76.26±4.81) had the highest scores, respectively, but major art consistently scored the lowest, which were 73.85±6.02, 74.97±5.53, and 72.59±4.84, respectively.” (lines 2-7, page 21, in Revised version).

References

 Ref.8: The citation is inaccurate.

Response: Revised.

 Ref. 11: I see no connection with the present study. Please correct.

Response: Replaced, Wang M. Physical Activity and Fitness of Children and Adolescents Aged 6-19 in China: Current Status and Trends [C]//Compilation of Abstract of Papers at the 10th National Sports Science Conference in 2015 (Part 1). 2015. (Ref 17, page 23, in Revised version).

 Ref.13: The citation is inaccurate.

Response: Revised.

 Ref.26: The citation is inaccurate.

Response: Deleted ref 26.

 Ref.41: The citation is inaccurate.

Response: Ref 41 is a Master thesis of Taiyuan University of Technology.

General Comment: Several references were either absent from online sources or were published exclusively in local journals. I strongly encourage authors to thoroughly examine all references, accurately document them, and exclusively utilize those from reputable academic journals

Response: Thoroughly reviewed all references and updated some of them from reproducible academic journals.

---

## [Decision Letter · Decision Letter 1]

9 Jul 2024

PONE-D-23-32339R1A comparative study on the physical fitness of college students from different grades and majors in Jiangxi provincePLOS ONE

Dear Dr. Sun,

Thank you for submitting your manuscript to PLOS ONE. After careful consideration, we feel that it has merit but does not fully meet PLOS ONE’s publication criteria as it currently stands. Therefore, we invite you to submit a revised version of the manuscript that addresses the points raised during the review process.

We look forward to receiving your revised manuscript.

Kind regards,

Miquel Vall-llosera Camps

Senior Staff Editor

PLOS ONE

Journal Requirements:

**Additional Editor Comments:**

Please address the following issues:

1) ". For example, among all majors, girls majoring in art tended to be slimmer. This high probability may be related to the difference between liberal humanities and science courses. There are more experimental courses in science and far more opportunities for exercise, while there are more liberal arts and history courses in humanities with longer sedentary hours"

This statement contradicts itself - it is saying girls majoring in art tend to be slimmer; but then says those in liberal arts courses have longer sedentary hours.

2) "Parents or students themselves do not attach importance to fitness."

This statement is not supported by evidence. Please provide the evidence or remove it.

3) " Constructing a healthy China requires the support of people's good exercise habits, especially for college students who are valuable talent and the future of our motherland."

Please rephrase the use of "motherland".

Reviewers' comments:

Reviewer's Responses to Questions

**Comments to the Author**

1. If the authors have adequately addressed your comments raised in a previous round of review and you feel that this manuscript is now acceptable for publication, you may indicate that here to bypass the “Comments to the Author” section, enter your conflict of interest statement in the “Confidential to Editor” section, and submit your "Accept" recommendation.

Reviewer #1: All comments have been addressed

2. Is the manuscript technically sound, and do the data support the conclusions?

Reviewer #1: Yes

3. Has the statistical analysis been performed appropriately and rigorously? 

Reviewer #1: Yes

4. Have the authors made all data underlying the findings in their manuscript fully available?

Reviewer #1: Yes

5. Is the manuscript presented in an intelligible fashion and written in standard English?

Reviewer #1: Yes

6. Review Comments to the Author

Reviewer #1: I would like to express my gratitude to the editor for giving me the opportunity to critically appraise this exceptional manuscript. I congratulate the authors on their diligent efforts. The manuscript has been considerably improved, and I have no further comments to make.

7. PLOS authors have the option to publish the peer review history of their article (what does this mean?). If published, this will include your full peer review and any attached files.

Reviewer #1: **Yes: **Dr. Mohamed Ahmed Said

---

## [Author Response · Author response to Decision Letter 1]

10 Jul 2024

1) "For example, among all majors, girls majoring in art tended to be slimmer. This high probability may be related to the difference between liberal humanities and science courses. There are more experimental courses in science and far more opportunities for exercise, while there are more liberal arts and history courses in humanities with longer sedentary hours"

This statement contradicts itself - it is saying girls majoring in art tend to be slimmer; but then says those in liberal arts courses have longer sedentary hours.

Response: Deleted the following sentence, "For example, among all majors, girls majoring in art tended to be slimmer" , to avoid ambiguity.

2) "Parents or students themselves do not attach importance to fitness."

This statement is not supported by evidence. Please provide the evidence or remove it.

Response: Deleted the following sentence, "Parents or students themselves do not attach importance to fitness."

3) " Constructing a healthy China requires the support of people's good exercise habits, especially for college students who are valuable talent and the future of our motherland."

Please rephrase the use of "motherland".

Response: Replaced "our motherland" with "China."

---

## [Editor Report · Decision Letter 2]

25 Jul 2024

A comparative study on the physical fitness of college students from different grades and majors in Jiangxi province

PONE-D-23-32339R2

Dear Dr. Sun,

We’re pleased to inform you that your manuscript has been judged scientifically suitable for publication and will be formally accepted for publication once it meets all outstanding technical requirements.

Kind regards,

Miquel Vall-llosera Camps

Senior Staff Editor

PLOS ONE
---

## [Editor Report · Acceptance letter]

12 Aug 2024

PONE-D-23-32339R2 

PLOS ONE

Dear Dr. Sun, 

I'm pleased to inform you that your manuscript has been deemed suitable for publication in PLOS ONE. Congratulations! Your manuscript is now being handed over to our production team.

Kind regards, 

on behalf of

Dr. Miquel Vall-llosera Camps 

Staff Editor

PLOS ONE